# Optimizing Crop Production with Bacterial Inputs: Insights into Chemical Dialogue between *Sphingomonas sediminicola* and *Pisum sativum*

**DOI:** 10.3390/microorganisms11071847

**Published:** 2023-07-21

**Authors:** Candice Mazoyon, Stéphane Firmin, Lamine Bensaddek, Audrey Pecourt, Amélie Chabot, Michel-Pierre Faucon, Vivien Sarazin, Fréderic Dubois, Jérôme Duclercq

**Affiliations:** 1Ecologie et Dynamique des Systèmes Anthropisés (EDYSAN, UMR7058 CNRS), Université de Picardie Jules Verne (UPJV), 80039 Amiens, France; candice.mazoyon@outlook.fr (C.M.); lamine.bensaddek@u-picardie.fr (L.B.); pecourta@agrostation.fr (A.P.); frederic.dubois@u-picardie.fr (F.D.); 2Agroécologie, Hydrogéochimie, Milieux et Ressources (AGHYLE, UP2018.C101) UniLaSalle, 60026 Beauvais, France; stephane.firmin@unilasalle.fr (S.F.); michel-pierre.faucon@unilasalle.fr (M.-P.F.); 3AgroStation, 68700 Aspach-le-Bas, France; sarazinv@agrostation.fr; 4UFR des Sciences, Université de Picardie Jules Verne (UPJV), 80039 Amiens, France; amelie.chabot@u-picardie.fr

**Keywords:** *Sphingomonas sediminicola*, pea, root exudates, plant–bacteria interaction, molecular dialogue plant–bacteria

## Abstract

The use of biological inputs is an interesting approach to optimize crop production and reduce the use of chemical inputs. Understanding the chemical communication between bacteria and plants is critical to optimizing this approach. Recently, we have shown that *Sphingomonas* (*S*.) *sediminicola* can improve both nitrogen supply and yield in pea. Here, we used biochemical methods and untargeted metabolomics to investigate the chemical dialog between *S. sediminicola* and pea. We also evaluated the metabolic capacities of *S. sediminicola* by metabolic profiling. Our results showed that peas release a wide range of hexoses, organic acids, and amino acids during their development, which can generally recruit and select fast-growing organisms. In the presence of *S. sediminicola*, a more specific pattern of these molecules took place, gradually adapting to the metabolic capabilities of the bacterium, especially for pentoses and flavonoids. In turn, *S. sediminicola* is able to produce several compounds involved in cell differentiation, biofilm formation, and quorum sensing to shape its environment, as well as several molecules that stimulate pea growth and plant defense mechanisms.

## 1. Introduction

Plant growth and productivity are closely associated with rhizospheric bacteria, such as plant growth-promoting rhizobacteria (PGPR), in a reciprocal relationship that benefits both partners [1,2]. These bacteria can live freely in the rhizosphere of plants or be directly associated with plants in a symbiotic relationship. Free PGPR contributes to plant health and development by producing plant growth regulators [3], increasing the availability of nutrients and water, and reducing the effects of diseases, pests, and environmental stressors. They can also increase the amount of soil organic matter, improve soil texture and structure, and promote micronutrient uptake [4,5,6]. Symbiotic PGPR such as those of the order *Hyphomicrobiales* (=*Rhizobiales*) are useful for agriculture because they are able to form nodules on the roots of legumes in which the atmospheric nitrogen (N_2_) fixed by the bacteria is converted into ammonia for the plant [7].

The interaction between plants and bacteria requires complex chemical communication, which can be viewed as a conversation between the two parties in which plants and bacteria exchange chemical signals to influence their growth and development [8,9]. First, plants and bacteria can sense each other’s chemical signals through specific receptors [10,11], which usually enables the recruitment of beneficial bacteria by the plant. Plants and bacteria can respond to chemical signals by altering their growth, metabolism, and functions that allow bacteria to colonize plant roots. This also affects the production of chemical signals, forming a feedback loop that leads to an equilibrium beneficial to both organisms [12,13]. PGPRs recruitment by plants requires a significant amount of energy to attract and select bacteria in the rhizosphere [14,15,16]. Plant roots release a variable and diverse set of compounds, including carbohydrates, amino acids, organic acids, and secondary metabolites [17,18,19]. The compounds released by the roots are an important nutrient source for soil bacteria and have an attractive effect [20]. In turn, PGPR produces compounds that promote plant growth and health [19,21]. Various studies have shown that the microbial community associated with the root can evolve depending on the chemical composition of the root exudates, which also depends on the plant species and its stage of development [11,19,22]. For effective interaction, it is important that the bacteria are metabolically adapted to the chemical signals released by the plant via its root exudates so that they can be recruited and establish themselves in the root system.

Agriculture based on the interaction between plants and bacteria promises to limit the use of chemical fertilizers that have negative effects on the environment and health acts [23,24]. Since the last decades, bacterial fertilizers containing free N_2_-fixing PGPR, such as *Pseudomonas stutzeri*, *P. oryzihabitans,* or *Azospirillum brasilense,* have been successfully used in agriculture as a strategy to improve plant growth in a sustainable way [6,25,26,27]. This aspect is even more pronounced in legumes such as pea, which is normally thought to interact with *Rhizobium* species [28]. Therefore, the molecular dialogs during this interaction have been studied in detail [29,30]. These interactions, which depend on the specific characteristics of the pea species and the bacterial species involved, make it possible to predict the effectiveness of the use of this bio-input. Despite the importance of this association, it is important to note that in conventionally plowed soils where peas are rotated, the bacterial community may also be predominantly influenced by other bacterial species, such as *Sphingomonas* [31,32,33,34].

Recently, we have shown that *Sphingomonas sediminicola* is also able to induce the formation of root nodules in peas and increase the production of plant biomass [35]. This new interaction raises the question of chemical communication between the two partners and offers the possibility of optimizing the effect of bacterial bio-input on a pea plant. Therefore, we examined a wide range of compounds, including carbohydrates, carboxylic acids, amino acids, polyphenols, and flavonoids in hydroponic pea cultures inoculated and non-inoculated with *S. sediminicola*. We compared these results with the metabolic phenotyping of *S. sediminicola* based on its ability to degrade various carbon, nitrogen, phosphorus, and sulfur sources. To get an overview of the effect of *S. sediminicola* on peas, we also analyzed the compounds released by the bacterium.

## 2. Materials and Methods

### 2.1. Plant and Bacteria Culture

*Sphingomonas sediminicola* (DSM-18106) was grown in R2A medium (VWR, Fontenay-sous-Bois, France) for 72 h at 30 °C under constant shaking at 150 rpm. Pea (*Pisum sativum*, cv. Douce Provence, Jardiland, Paris, France) seeds were surface-sterilized with a 3.5% (*v*/*v*) bleach solution, cold-stratified for 48 h, and germinated in the dark on a 1% (*w*/*v*) agar medium at 21 °C. Five days after germination, etiolated seedlings were transferred to hydroponic system.

### 2.2. Hydroponic Growth Conditions

Peas were cultivated in a homemade hydroponic system using an 8-L plastic pot filled with 6 L of sterile distilled water supplemented with 0.23 g L^−1^ of Murashige and Skoog’s basal salt mixture [36]. Each device had an air pump with a flow rate of 78 L h^−1^ to maintain the oxygen supply. Five plants were arranged per system. Substances released via pea root exudates were collected at the emergence of the first leave, second internode, and flowering by withdrawing 1 L of hydroponic solution at each stage and replacing it with 1 L of hydroponic solution. Half of the devices were inoculated with *S. sediminicola* to obtain a concentration of 10^6^ CFU L^−1^.

### 2.3. Extraction and Analysis of Hydroponic Compounds

Samples of the hydroponic solution were filtered with Whatman paper (GE Healthcare, Chicago, IL, USA), and the chemical compounds in these solutions were selectively absorbed by Amberlite XAD-8 resin (Sigma-Aldrich, Taufkirchen, Germany), loaded onto the Chromabond C-18 SPE columns (Macherey-Nagel, Düren, Germany), dropwise desorbed with pure methanol then concentrated using a rotary evaporator at 40 °C.

An ultra-high performance liquid chromatography (UPLC) system (Waters Corp., Milford, MA, USA) was used for global analysis of the concentrated samples using MassLynx^TM^ software (v4.2, Waters Corp., Milford, MA, USA). Global detection of biological information was performed using high-throughput G2Si High-definition mass spectrometry (Waters Q-TOF SYNAPT™, Waters Corp., Milford, MA, USA). Mobile phases consisted of a gradient of 0.1% (*v*/*v*) formic acid in Milli-Q water (A) and MeOH (B). The flow rate was maintained at 0.4 mL min^−1^. Elution was performed at a gradient of 10% B for 1 min, then increased to 90% over 6 min and held for 3 min. Initial conditions were then restored over 1 min and held for 3 min for re-equilibration.

Total polyphenol content was determined spectrometrically by the Folin–Ciocalteu method [37] using tannic acid in the range of 0–100 μg mL^−1^ as a reference standard. The flavonoid assay [38] was performed using quercetol (0.05 mg mL^−1^) as the reference standard. D-glucose, D-fructose, and sucrose were determined enzymatically using the sucrose/D-glucose kit (R-Biopharm, Pfungstadt, Germany). D-xylose, L-arabinose, D-galactose, xylitol, and D-sorbitol were determined using specific enzymatic kits (Megazyme Assay Kits, Megazyme International Ireland Limited, Wicklow, Ireland).

For the determination of amino acid content, hydroponic samples were concentrated 3-fold by ultrafiltration using a polyethersulfone membrane with a cut-off of 3 kDa (Vivaspsin^®^ 2 concentrator). The identification and quantification of amino acids were performed using a Kromasil C18—100 Ǻ-5 μm 4.6 × 250 mm column after derivation with o-phthalaldehyde (OPA). For derivatization, 50 μL of sample was added to 50 μL of borate buffer containing 0.25% (*w*/*v*) OPA and 5% (*v*/*v*) β-mercaptoethanol. An injection of 5 μL of this solution was performed. The proline content was determined using a ninhydrin assay [39].

Organic acid contents were determined by liquid chromatography with UV detector (HPLC-UV) method [40]. After centrifugation at 12,000 rpm for 2 min, the supernatant of hydroponic samples was filtered to 0.2 μm, and 50 μL of the filtrate was injected into the analytical system equipped with a C18 column (Luna, 5 μm Phenomenex). Organic acids were separated in an isocratic medium with a mobile phase consisting of 25 mM phosphate buffer and methanol (99/1, *v*/*v*; pH 2.4). The detection of carboxylic acids was performed at 210 nm.

### 2.4. Microarray Plates for Phenotypic Characterization of S. sediminicola

We used pre-configured Biolog Phenotype MicroArrays (Biolog Inc., Harvard, CA, USA) to phenotype the metabolic potential of *S. sediminicola* against 190 carbon (PM1 and PM2), 95 nitrogen (PM3), 59 phosphorus, and 35 sulfur (PM4) sources. According to the manufacturer’s instructions, bacteria were grown on an R2A agar plate, and colonies were harvested to achieve the required transmittance and mixed with a buffer containing IF-0 concentration 1.2X (Biolog Inc.) and the redox dye tetrazolium violet. For PM3 and PM4, 125 μL 2 M sodium succinate/200 μM ferric citrate was added to the mixture to regulate the pH of the suspension. Each plate was incubated in an Omnilog (Biolog Inc.) at 25 °C for 96 h. Three biological replicates per plate were performed. The phenotypic response of *S. sediminicola* to PM substrates was monitored by the color change in each well. Reduction of tetrazolium salt by bacterial dehydrogenases and reductases results in a purple formazan dye. Therefore, a color reaction indicates that the bacteria are actively metabolizing a substrate in the well, while the absence of a color change means that the bacteria cannot metabolize the substrate. The rate of color change in each well was monitored every 15 min at 490 nm and stored in Omnilog units generated by Biolog Data Analysis Software (v1.7, Biolog Inc.). For each source, a color intensity greater than 500 Omnilog units were considered positive in at least two of three replicates.

### 2.5. Untargeted Metabolomics of Sphingomonas sediminicola Culture Medium

The analysis was performed with the supernatant obtained by centrifugation of a culture medium in which *Sphingomonas sediminicola* had been grown, reaching a concentration of 10^6^ CFU mL^−1^. This untargeted metabolomic analysis was performed by Beijing Genomics Institute (BGI, Tai Po, Hong Kong). The culture was grown in R2A medium, according to the method described by Dunn et al. [41]. Metabolite extracts were analyzed on a Waters 2D UPLC (Waters, Milford, MA, USA), coupled to a Q-Exactive mass spectrometer (Thermo Fisher Scientific, Waltham, MA, USA) with a heated electrospray ionization (HESI) source and controlled by the Xcalibur 2.3 software program (Thermo Fisher Scientific, Waltham, MA, USA). Chromatographic separation was performed on a Waters ACQUITY UPLC BEH Amide column (1.7 μm, 2.1 mm × 100 mm, Waters, Milford, MA, USA) at 30 °C.

For the untargeted metabolomics data, the mass spectrometry raw data collected by LC-MS/MS were imported into Compound Discoverer 3.1 (Thermo Fisher Scientific, Waltham, MA, USA) for data processing. The identification of metabolites is a combined result of BMDB (BGI Metabolome Database), mzCloud, and ChemSpider (HMDB, KEGG, LipidMaps) databases. Main parameters of metabolite identification: Precursor Mass Tolerance < 5 ppm, Fragment Mass Tolerance < 10 ppm, RT Tolerance < 0.2 min. The results of the Compound Discoverer 3.1 export were imported into metaX [42] for data preprocessing. A total of 4563 and 1851 compounds were detected in the positive (pos) and negative (neg) modes, respectively. After preprocessing, 1487 (pos) and 869 (neg) of these compounds were identified.

### 2.6. Data Analysis

To compare the composition of the compounds present in the hydroponic solutions, we used the R package *agricolae* [43] and a Kruskal–Wallis nonparametric one-way analysis of variance followed by a Conover–Iman post hoc test if significant and Holm’s *p*-adjust method for multiple comparisons.

Differential metabolites screening conditions were performed with a Fold-Change (FC) ≥ 1.2 or ≤0.83 and a *p*-value < 0.05 obtained through Student’s *t*-test and corrected for false discovery rate (FDR). Cluster analysis of differential metabolites was performed with log_2_ transformation and z-score normalization (zero-mean normalization). The clustering algorithm used hierarchical cluster based on Euclidean distance. In the bubble plots for pathway enrichment analysis, the enrichment factor (RichFactor) on the *X*-axis is the number of differential metabolites annotated for the pathway divided by all identified metabolites annotated for the pathway. The larger the value, the greater the proportion of differential metabolites annotated for the pathway. The dot size represents the number of differential metabolites annotated for that metabolic pathway. Volcano plot based on log_2_FC and the Heatmap of the differential metabolites were generated using the R packages. All statistical analyzes were performed using R software (v4.2.2, The R Foundation, https://www.r-project.org/, accessed on 13 February 2023).

## 3. Results

### 3.1. The Composition of the Hydroponic Solution Is Influenced by S. sediminicola and the Stage of Pea Development

The compounds present in the hydroponic solution samples were structured around two axes of PCA ordination, which accounted for 60% of the variance between samples (Figure 1). Overall, axis 1 is related to bacterial inoculation, while axis 2 is related to the stage of pea development.

At the first leaf emergence stage, the composition of the hydroponic solution was very poor in polyphenols and flavonoids (Appendix A). Serine, glycine, and alanine were the major amino acids. Citrate and especially acetate characterized the organic acid content, while fructose, glucose, and sucrose determined the carbohydrate content. Between the emergence of the first leaf and the formation of the second internode, there was little change in the composition of the hydroponic solution. However, a change was observed in carbohydrates, with a sharp increase in fructose content and a decrease in sorbitol and xylitol content. At the flowering phase, the solutions were richer in tryptophan and especially in cysteine, which becomes the most abundant amino acid in the solutions. Additionally, citrate and fumarate also showed an increase during this phase, although not to the same extent as lactate, which reached levels similar to acetate. The content of carbohydrates also evolved, with fructose remaining proportionally dominant and an increase in the content of galactose (Appendix A).

Inoculation of peas with *S. sediminicola* resulted in a hydroponic solution whose composition was almost the same as that of the non-inoculated peas until the emergence of the first leaf (Figure 1, Appendix A). However, this inoculation resulted in higher flavonoid, galactose, glucose, and xylitol contents than under the non-inoculated conditions. Between the emergence of the first leaf and the formation of the second internode, the composition of the hydroponic solutions of the inoculated peas evolved with an increase in the content of many amino acids. The content of polyphenols and flavonoids remained the same between the two stages, as did the content of carbohydrates. However, the content of glucose increased, while the content of galactose decreased (Appendix A). The composition of the hydroponic solution remained relatively stable between the formation of the second internode and flowering of the inoculated peas, with consistently higher levels of amino acids, flavonoids, and polyphenols than in the non-inoculated condition. UPLC analysis allowed the identification of different terpenoids and flavonoids in the solution, including pisatin (Appendix A and Appendix A). A clear difference between non-inoculated and inoculated conditions was also observed for other compounds. This was true for the lactate and galactose content, which were very low in the flowering stage of the inoculated peas.

### 3.2. S. sediminicola Had a Specialized Carbon Metabolism and a Generalist Nitrogen Metabolism

To determine whether the compounds released in root exudates can be metabolized by *S. sediminicola*, we examined the metabolic capacities of the bacteria (Figure 2; Appendix A). Of the 333 substrates tested, 189 were degraded by *S. sediminicola* (>500 Omnilog Unit, OU), especially amino acids (55 of 77), nucleic acids/nucleotides/nucleosides (31/40), and some specific nitrogen sources such as nitrogen dioxide, nitrate, nitrite, and urea (Figure 2, Appendix A). *S. sediminicola* showed a strong ability to metabolize certain pentoses such as L-lyxose (12116 OU), D-xylose (6753 OU), D-ribose (7269 OU), L-arabinose (8509 OU), in contrast to numerous hexoses such as D-glucose (664 OU), D-fructose (652 OU), D-mannose (842 OU), or D-galactose (634 OU). Similarly, di- and trisaccharides were degraded only very slightly or not at all. Carboxylic acids form a group of substrates that are only weakly degraded by the bacteria but with a preference for 5-keto-D-gluconic acid (7482 OU), L-tartaric acid (3834 OU), oxalomalinic acid (1882 OU), and for two aminocarboxylic acids, D,L-α-aminocaprylic acid (7369 OU) and ε-amino-N-caproic acid (10123 OU).

### 3.3. Untargeted Metabolomics

Untargeted metabolomic analysis revealed that there are many metabolites whose levels changed drastically during *S. sediminicola* growth (Figure 3a,b and Appendix A). Indeed, the levels of 174 and 352 compounds identified in the positive and negative modes, respectively, decreased during *S. sediminicola* growth (Figure 3c,d and Appendix A). Among them were many amino acids (e.g., arginine, asparagine, aspartic acid, glutamic acid, lysine, methionine, phenylalanine, serine, and tryptophan), peptides, or carbohydrates such as ribose, mannose, and threose (Appendix A). The levels of 389 (pos) and 316 (neg) identified molecules, including iminoquinoline, p-coumaroyl-homoserine lactone (pC-HSL) and 2-amino-glucopyranosyl mannitol, increased during bacterial growth. Additionally, higher contents of indole-3-carboxylic acid, methylimidazoleacetic acid, and indole-3-acetic acid were detected after *S. sediminicola* growth. 9-(α-D-glucosyl)kinetin and kinetin were also found in this condition. Furthermore, N-acetylsphingosine, galactopinitol b, 8-methoxykynurenic acid, and monoglycosyl-N-acylsphingosine were present in the *S. sediminicola* medium culture. Overall, the bacteria caused an accumulation of compounds involved in amino acids, pentoses, and secondary metabolic pathways (Figure 3e,f).

## 4. Discussion

Plants release root exudates [44,45] to influence soil bacterial communities and attract certain bacteria to the rhizosphere [46,47]. However, the changes in root exudate composition induced by bacteria involved in plant growth remain poorly understood.

During pea development, carbohydrates, particularly hexoses and sucrose, accumulate in the hydroponic solution. Fructose levels notably increase during the transition from leaf emergence to flowering, while sorbitol and xylitol become undetectable. These compounds, being simple sugars, can serve as a carbon and energy source for many bacteria [48]. Their presence in the hydroponic solution may serve as a chemical communication strategy to recruit bacterial partners capable of utilizing labile carbon. Notably, these molecules have been described to effectively attract bacterial species such as *Bacillus*, *Methylobacterium,* or *Pseudomonas* [22]. In addition, the abrupt increase in galactose content in pea hydroponic solution at flowering is consistent with Knee et al. [49] but also with the chemotaxic character of the molecule. Indeed, this compound induces a chemotaxis effect on some PGPR, such as *Pseudomonas* with legume roots or *Bacillus velezensis* with cucumber roots [49,50].

In the hydroponic solution of peas inoculated with *S. sediminicola*, hexose contents followed similar trends until the formation of the second internode. Compared with the non-inoculated peas, higher pentose, and lower galactose contents were observed under the conditions with *S. sediminicola*. This indicates carbohydrate communication adapted to *S. sediminicola*, which fits with the more pronounced metabolic preference of the bacterium for pentoses compared to hexoses and the predominant use of these compounds in its culture medium. Pentoses, like hexoses, are labile carbon sources, but their degradation may be more complex than that of hexoses because the enzymes required for their degradation may be less abundant in soil microorganisms [51]. In addition, pentoses may be incorporated into complex polymers such as cellulose and hemicellulose, which are important components of fresh soil organic matter [52]. In this case, their degradation may be slower and require the activity of specialized microorganisms such as *S. sediminicola* [32].

The hydroponic solution of non-inoculated peas contains high concentrations of organic acids, particularly acetic, citric, lactic, and furamic acids. Acetic and citric acids are generally released by plants to increase nutrient uptake, such as phosphorus [53], manganese [54], iron, and zinc [55]; stimulate biofilm production and motility of some PGPR [56,57,58]; induce nitrogen-fixing bacteria [59], *Rhizobium* IC3109 [58] and acid-forming bacteria (*Acetobacter* and *Gluconacetobacter* [60]); or stimulate bacteria such as *Pseudomonas*, which is capable of producing plant growth hormones and protecting plants from disease [61]. Citric acid has also been shown to recruit phosphate-solubilizing bacteria (*Pseudomonas putida* [62]) or symbiotic bacteria (*Burkholderia cepacia* and *Rhizobium leguminosarum* [58,63,64,65]). Similarly, lactic acid released from the roots of legumes has been shown to attract bacteria such as *Lactobacillus* and *Pediococcus* [66,67]. This organic acid is also known for its antimicrobial properties, which can help the plant against pathogens in the rhizosphere [68,69]. Furamic acid was also present in the hydroponic solution. This acid plays an important role in recruiting plant-friendly rhizobacteria, symbiotic nitrogen-fixing bacteria, phosphate-solubilizing bacteria, or bacteria that produce antimicrobial compounds that protect plants from soil pathogens [61,70,71]. Therefore, pea releases a wide range of organic acids that serve as a broad-spectrum attractant for rhizosphere bacteria to entice any bacteria capable of perceiving and metabolizing these compounds, potentially including beneficial bacteria that promote plant growth. In the presence of *S. sediminicola*, the hydroponic solution of peas also contains the same organic acids, except for lactic acid. The presence of these organic acids in the hydroponic solution of peas inoculated with *S. sediminicola* is consistent with the metabolism of the bacteria, which can utilize acetic and furamic acids but not lactic acid. (Appendix A). Other organic acids, such as tartaric acid and butyric acid, which are highly metabolized by *S. sediminicola* and involved in regulating plant–microorganism communication, could be further quantified [63,72].

In the later growth stages, plants secrete amino acids and polyphenols, which shape bacterial communities [18,73]. Amino acids serve as a nitrogen source for bacteria [74], leading to competition for nutrient sources among them [75]. Polyphenols and flavonoids also act as nutrient sources and chemoattractants for bacteria involved in infections or symbiotic relationships, such as *Agrobacterium tumefaciens* or rhizobia [73,76]. Previous studies have shown that *Pseudomonas fluorescens* and *P. aeruginosa* induce the secretion of specific phenolic acids and increase the total content of polyphenols at different plant growth stages, especially in advanced stages of chickpea plants [77]. Our study indicates that the hydroponic solution of peas inoculated with *S. sediminicola* is enriched with amino acids and polyphenols from the formation of the second internode to flowering. This suggests that these compounds are a response of the peas to the presence of the bacteria and may affect the bacterial community structure. In addition, bacteria capable of metabolizing a wide range of amino acids from root exudates have a selective advantage in the plant rhizosphere [61,78]. The metabolism of *S. sediminicola* exhibits a high affinity for amino acid substrates and degrades nearly 80% of all such substrates, which also explains the large number of amino acids used by the bacteria in its culture medium.

Flavonoids are plant secondary metabolites released into the rhizosphere, playing a crucial role in chemical communication with rhizosphere bacteria [79]. They act as chemoattractants for bacteria involved in symbiotic interactions with the plant, such as rhizobia [80]. Additionally, flavonoids stimulate biofilm production, promoting bacterial colonization of roots, which can lead to better plant growth and health [81]. Interestingly, the hydroponic solution of peas inoculated with *S. sediminicola* contained flavonoids, which were not detected in the non-inoculated condition. Moreover, flavonoid content increased with pea development. Some identified flavonoids in the solution corresponded to pisatin, a plant phytoalexin with an antimicrobial activity produced by plants in response to infection or stress [82,83,84]. Thus, the induction of pisatin production in the presence of *S. sediminicola* can be considered a pea defense response. It is noteworthy that *S. sediminicola* has the ability to form nodules on pea roots during nitrogen stress, aligning with observations in pea-*Rhizobium* interaction and the regulatory role of isoflavonoids in nodulation [35,85,86]. Fragmentation analysis of other peaks from biochemical analysis would help in the identification of other flavonoid compounds.

During their growth, rhizospheric bacteria produce various molecules for their own growth, including galactoside [87]. Galactopinitol b was among the galactosides detected in the culture medium after the growth of *S. sediminicola*. These galactosides serve as carbon and energy sources for the bacteria, facilitating their growth in the rhizosphere. In addition, certain rhizobia can synthesize galactopinitol b, which may play a role in establishing symbiotic relationships with plants [88,89,90]. In the culture medium, we also detected the presence of various components involved in cell differentiation, biofilm formation, and quorum sensing. Thus, the bacterial culture medium was enriched with methyl-4-hydroxy-6-methyl-2-pyrone as well as p-coumaroyl-homoserine lactone (pC-HSL). These compounds are quorum-sensing molecules, facilitating communication and coordination among bacteria to regulate their collective behavior [91,92]. In the same manner, monoglycosyl-N-acylsphingosine, which is involved in biofilm formation, may also regulate plant growth and modulate the plant immune system [93,94]. Interestingly, enrichment of the R2A medium with auxin-associated molecules (indole-3-carboxylic acid, methylimidazoleacetic acid, and indole-3-acetic acid) was also observed. Therefore, *S. sediminicola* might affect plant growth and development and auxin homeostasis in roots via these molecules [95]. Thus, it would be interesting to investigate whether the presence of *S. sediminicola* at the root level affects plant development, root system architecture, plant defense mechanisms, but also the organization of microbial communities in the rhizosphere.

Overall, we provide new information on how pea plants modulate the composition of their root exudates to recruit bacteria from which they, and *Sphingomonas sediminicola* in particular, can benefit. We also describe for the first time the full metabolic and catabolic potential of this bacterium. In future experiments, the inclusion of other rhizobia would provide valuable comparative data on the effects of different bacterial species. Such information would not only improve our understanding of the specific interactions between these bacteria and pea plants but also shed light on the broader context of plant–microbe associations in agricultural systems. It is important to note that our study was conducted under controlled hydroponic conditions that allowed precise control of nutrient availability and environmental factors. However, it would be equally interesting to study the chemical dialog between *S. sediminicola* and pea plants in an agricultural soil context. Soil conditions have a significant impact on microbial community structure and nutrient dynamics, which could influence the nature and extent of plant–microbe interactions. Future studies in agricultural soil systems would provide a better understanding of the practical implications and transferability of our results to real agricultural conditions.

## 5. Conclusions

The chemical dialog underlying plant–microbe interactions are essential to understanding the use of microorganisms in agriculture. We have shown that pea plants employ a strategic mechanism to selectively recruit specific bacteria to their rhizosphere that have the potential to confer benefits to the plants. During its development, the plant modulates its root exudation by first releasing various generalist molecules that allow recruitment focused on fast-growing bacteria. Then, the exudation evolves further by releasing more specific compounds that are tuned to the metabolic potential of the targeted bacterial microbiota, such as *Sphingomonas sediminicola*. In return, the bacteria provide various services to the plant that allow it to develop better and cope with environmental conditions [35].

This beneficial interaction between the plant and a bacterium is a step towards understanding the interaction between plants and bacteria. Indeed, the plant in agricultural soil is in relationship with a large variety of organisms, some of which are beneficial, others less so. Therefore, it is important to understand how the plant adapts its recruitment strategies to select the most important partners. It is also important to identify the mechanisms that enable molecular and chemical dialog between partners. Understanding all these strategies is potentially an important lever for implementing more sustainable agriculture.

## Figures and Tables

**Figure 1 microorganisms-11-01847-f001:**
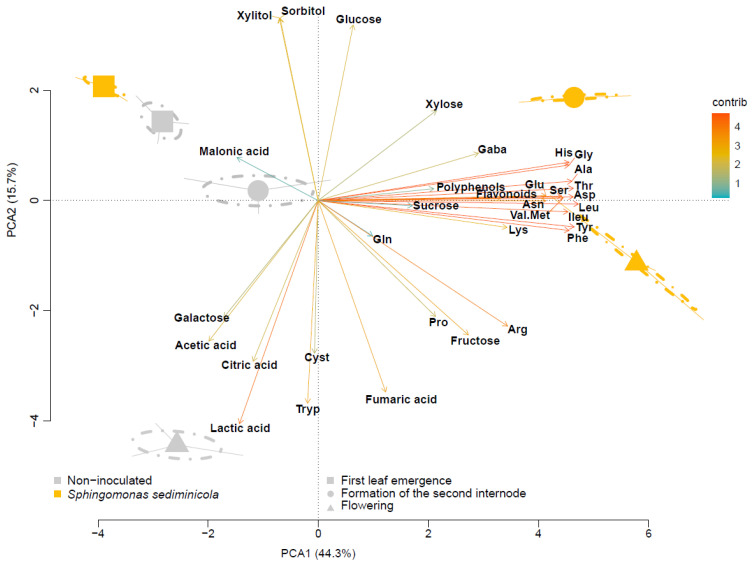
Principal component analysis (PCA) of polyphenol, flavonoid, amino acid, and carbohydrate content in hydroponic solutions of peas inoculated or not with *Sphingomonas sediminicola* at the time of first leaf emergence, second internode, and flowering.

**Figure 2 microorganisms-11-01847-f002:**
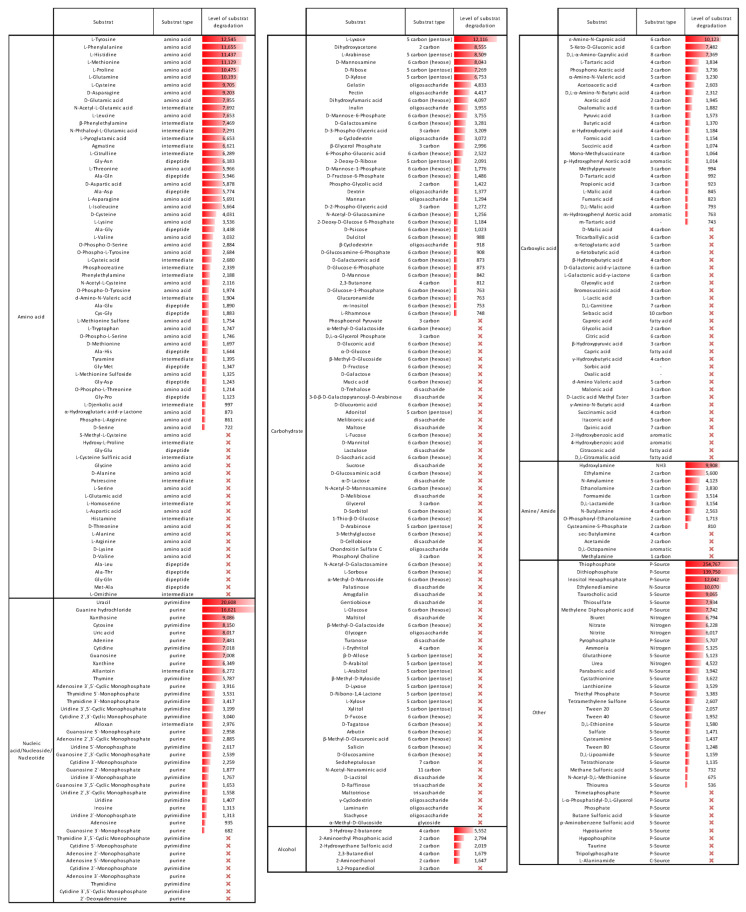
Metabolic potential of *S. sediminicola* against PM substrates. For each source, the level of substrate degradation, expressed in Omnilog units, was considered only if it reached a minimum value of 500 Omnilog units in at least two of the three replicates. For degraded substrates, the value given is the average of the three replicates; otherwise, a red cross indicates that the substrate was not degraded by *S. sediminicola*.

**Figure 3 microorganisms-11-01847-f003:**
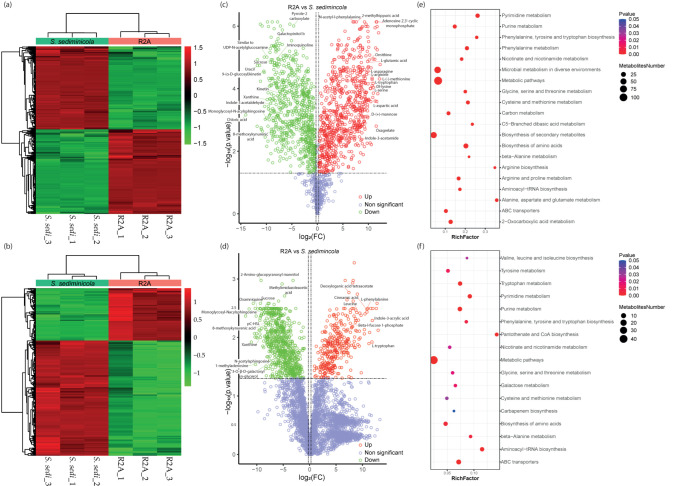
Untargeted metabolomics of *Sphingomonas sediminicola* in R2A medium. (**a**,**b**) Heatmap of differential metabolites in positive (**a**) and negative (**b**) ion modes. Each row corresponds to a differentially expressed metabolite, while each column represents a specific sample. The color gradient, ranging from green to red, indicates the abundance level of the differentially expressed metabolites, with green representing low abundance and red representing high abundance. (**c**,**d**) Volcano plot of differential metabolites in positive (**a**) and negative (**b**) ion modes. Each point represents a metabolite, horizontal coordinates indicate different multiplicities of differential metabolites (log_2_ values), vertical coordinates indicate *p*-values (−log_10_ values), grey indicates metabolites with no significant differences, red indicates up-regulated metabolites (Up), and green indicates down-regulated metabolites (Down). (**e**,**f**) bubble plots of metabolic pathway enrichment in positive (**e**) and negative (**f**) ion modes. The x-axis in the figure represents the ratio of differentiated metabolites in a specific metabolic pathway to the total number of identified metabolites within that pathway (RichFactor). A higher value indicates a greater enrichment of differential metabolites in the pathway. The color of the dots corresponds to the *p*-value obtained from the hypergeometric test. A smaller *p*-value indicates a more reliable test. The size of the dots represents the number of differential metabolites present in the respective metabolic pathway.

## Data Availability

Not Applicable.

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
