# Peer review of "Optimizing Crop Production with Bacterial Inputs: Insights into Chemical Dialogue between Sphingomonas sediminicola and Pisum sativum"

_microorganisms, 2023, doi:10.3390/microorganisms11071847_

Round 1

Reviewer 1 Report

Sphingomonas sediminicola can improve both nitrogen supply and yield in pea. Authors o investigated the chemical dialog between S. sediminicola and pea by measuring the metabolites of these interacting partner. The results contribute to the application of plant growth-promoting Rhizobacteria in agricultural production.

Why don't authors design an experiment with Rhizobia and S. sediminicola  to determine various metabolites? Since S. sediminicola helps Pea improve  nitrogen supply?

Additionally, are there any differences in metabolite secretion between hydroponic and soil cultures?

Minor editing of English language required.

Author Response

Point 1: Why don't authors design an experiment with Rhizobia and S. sediminicola to determine various metabolites? Since S. sediminicola helps Pea improve nitrogen supply?

Response 1:. Thank you for your valuable feedback on our submitted manuscript. We appreciate your suggestion to include Rhizobium in our study. We are aware that studying the effects of different bacterial species, such as Rhizobium and S. sediminicola, would provide a more comprehensive understanding of plant-microbe interactions and their impact on agricultural practices. In our current study, we focused on exploring the chemical dialog between S. sediminicola and pea plants. Given the limited information available on Sphingomonas species, we believe it was important to focus on this specific bacterium as a starting point. However, we also believe that the inclusion of Rhizobium in our experimental design would allow a direct comparison of their respective metabolic profiles and nitrogen supply capabilities. We plan to conduct further studies to investigate these important aspects. We also plan to investigate the compatibility of Rhizobium and S. sediminicola and explore the combined effects of these two bacterial species on metabolite profiles and nitrogen supply of pea plants. We believe that these additional experiments will greatly expand the scope and applicability of our study.

Point 2: Are there any differences in metabolite secretion between hydroponic and soil cultures?.

Response 2: We appreciate your suggestion to investigate the differences in root exudation between hydroponics and soil cultures. This comparison would provide valuable insights into the influence of growth conditions on plant-microbe interactions and metabolite dynamics. We therefore took these elements into account at the end of the discussion.

Reviewer 2 Report

I have read the paper carefully and find the topic is interesting. Under the hypotesis that understanding the chemical communication between bacteria and plants is critical to optimizing this approach, the authors examined a wide range of compounds, including carbohydrates, carboxylic acids, amino acids, polyphenols, and flavonoids in hydroponic pea cultures inoculated and non-inoculated with S. sediminicola. Formally, the article is well structured. The introduction provides sufficient information to understand the objectives. Material and methods are well described. The results and conclusions are consistent with the proposed objectives, but it need to be tested with real data for implementing more sustainable agriculture. I only have a few comments and suggestions that authors must consider before publication:

-2.3. Extraction and analysis of hydroponic compounds. If possible shorten, please

-The same for: 2.5. Untargeted Metabolomics of Sphingomonas sediminicola culture médium

-I am not sure if figure 2 is nerccesary. In any case it doesn't look right

- The same for figure 3

- Reduce the length of the discussion to the order of 10%

- I suggest to introduce some experimental photos

- Finally, the authors should clearly indicate the scientific novelty of the described research. When analyzing the literature, they should indicate what elements of their research contribute to knowledge.

Author Response

Point 1: -2.3. Extraction and analysis of hydroponic compounds. If possible shorten, please.

Response 1: We admit that we originally provided a large amount of information in this section. We have made an effort to condense it while ensuring that the most important elements necessary for understanding the experimental protocol are preserved.

Point 2: -The same for: 2.5. Untargeted Metabolomics of Sphingomonas sediminicola culture medium.

Response 2: This section has also been reduced.

Point 3:                 - I am not sure if figure 2 is nerccesary. In any case it doesn't look right

                               - The same for figure 3.

Response 3: Figures 2 and 3 are important for understanding the article because they show which substrates are metabolized by Sphingomonas sediminicola and give an idea of the intensity of this degradation (Figure 2) and the metabolites produced by the bacterium in its culture medium (Figure 3). If we delete them, the reading of the article becomes rather complicated and they have to be replaced by very long and cumbersome tables, but they can be found in the supplementary data. Based on feedback from another reviewer, I think the problem is with the quality of the illustrations when the file was compiled. The illustrations were supplied in 600 dpi. There was also a version of the illustrations in PDF format with a very high resolution.

I tried to improve the Figures, which you can find individually in the Mazoyon et al_Figures.rar archive.

Point 4: Reduce the length of the discussion to the order of 10%.

Response 2: We have shortened the discussion but added a new paragraph to discuss the fact that our results were obtained under hydroponic conditions and future studies conducted in agricultural soil systems would therefore help to broaden our understanding of the practical implications and transferability of our results to real agricultural conditions.

Point 5: I suggest to introduce some experimental photos

Response 5: We appreciate the reviewer's suggestion to include experimental photos in our study. We understand the value of visual documentation in presenting experimental setups and results. However, we would like to clarify that the primary focus of our study was to investigate chemical interactions and perform biochemical analyzes, not visual observations. Therefore, we chose not to include experimental photos as they may not provide significant additional information or enhance the scientific understanding of the phenomena studied.

Instead, we made an effort to fully describe our experimental procedures, methods, and analysis techniques. Our goal was to provide transparency and allow other researchers to accurately replicate our work. We believe that the textual descriptions, along with the data and results presented, effectively convey the experimental design and results of our study.

Point 6: Finally, the authors should clearly indicate the scientific novelty of the described research. When analyzing the literature, they should indicate what elements of their research contribute to knowledge.

Response 6: We have tried to make the innovative aspect of our study clearer in the discussion, especially at the end.

Reviewer 3 Report

The topic of the manuscript entitled "Optimizing crop production with bacterial inputs: insights into chemical dialogue between Sphingomonas sediminicola and Pisum sativum" the subject is in line with the scientific profile of the journal "Microorganisms". The graphic part presenting the results requires improvement, i.e. Metabolic potential of S. sediminicola against PM substrates presented in Figure 2 is illegible. Also the quality of Figure 3c,d,e,f should be improved. In my opinion, the chapter "Introduction, Material and methods, Discussion, Conclusions and Abstract" were presented correctly and I have no comments to them.

Round 2

Reviewer 2 Report

I went through the revised version of the manuscript and found that it had improved from the original manuscript. Therefore, the article is suitable for publication.